# $(p,q)$-Hermite–Hadamard Inequalities for Double Integral and $(p,q)$-Differentiable Convex Functions

**Julalak Prabseang [1], Kamsing Nonlaopon [1],\* and Jessada Tariboon [2]**

[1] Department of Mathematics, Faculty of Science, Khon Kaen University, Khon Kaen 40002, Thailand; julalak.pra@kkumail.com

[2] Department of Mathematics, Faculty of Applied Science, King Mongkut's University of Technology North Bangkok, Bangkok 10800, Thailand; jessada.t@sci.kmutnb.ac.th

**\*** Correspondence: nkamsi@kku.ac.th; Tel.: +668-6642-1582

**Abstract:** The aim of this paper is to establish some new $(p,q)$-calculus of Hermite–Hadamard inequalities for the double integral and refinements of the Hermite–Hadamard inequality for $(p,q)$-differentiable convex functions.

**Keywords:** Hermite–Hadamard inequalities; $(p,q)$-derivative; $(p,q)$-integral; convex functions

## 1. Introduction

Quantum calculus is the study of calculus without limits and is sometimes called $q$-calculus. In $q$-calculus, we obtain the original mathematical formulas when $q$ tends to one. The beginning of the study of $q$-calculus can be dated back to the era of Euler (1707–1783), who first launched the $q$-calculus in the tracks of Newton's work on infinite series. Then, in the early Twentieth Century, Jackson [1] defined an integral, which is known as the $q$-Jackson integral, and studied it in a systematic way. The subject of $q$-calculus has many applications in the field of mathematics and other areas such as number theory, special functions, combinatorics, basic hypergeometric functions, orthogonal polynomials, quantum theory, mechanics, and the theory of relativity and physics. In recent years, the topic of $q$-calculus has increasingly interested many researchers. For more details, see [2–9] and the references therein. Recently, Tunç and Göv [10–12] studied the concept of $(p,q)$-calculus over the intervals of $[a,b] \subset \mathbb{R}$. The $(p,q)$-derivative and $(p,q)$-integral were defined and some basic properties are given. Furthermore, they obtained some new result for the $(p,q)$-calculus of several important integral inequalities. Currently, the $(p,q)$-calculus is being investigated extensively by many researchers, and a variety of new results can be found in the literature [13–18] and the references cited therein.

Mathematical inequalities are important to the study of mathematics, as well as in other area of mathematics such as analysis, differential equations, geometry, etcetera.

In 1893, Hadamard [19] investigated one of the fundamental inequalities in analysis as:

$$f\left(\frac{a+b}{2}\right) \leq \frac{1}{b-a}\int_a^b f(x)dx \leq \frac{f(a)+f(b)}{2}, \tag{1}$$

which is now known as the Hermite–Hadamard inequality.

In 2014, Tariboon and Ntouyas [20] studied the extension to $q$-calculus on the finite interval of (1), which is called the $q$-Hermite–Hadamard inequality, and some important inequalities. Next, Alp et al. [21] approved the $q$-Hermite–Hadamard inequality and then obtained generalized $q$-Hermite–Hadamard inequalities.

In 2018, Mehmet Kunt et al. [22] proved the left-hand side of the $(p,q)$-Hermite–Hadamard's inequality of (1) through $(p,q)$-differentiable convex and quasi-convex functions, and then, they gave some new $(p,q)$-Hermite–Hadamard's inequalities.

In 2019, Prabseang et al. [23] established the $q$-calculus of Hermite–Hadamard inequalities for the double integral as:

$$f\left(\frac{a+b}{2}\right) \leq \frac{1}{(b-a)^2} \int_a^b \int_a^b f(tx+(1-t)y)dxdy \leq \frac{f(a)+f(b)}{2}, \tag{2}$$

which was given by Dragomir [24]. Moreover, they obtained refinements of the Hermite–Hadamard inequality for $q$-differentiable convex functions.

The aim of this paper is to present the $(p,q)$-calculus of Hermite–Hadamard inequalities for double integrals (2) and refinements of the Hermite–Hadamard inequality. These are obtained as special cases when $p=1$ and $q \to 1$.

Before we proceed to our main theorem, the following definitions and some concepts require some clarifications.

## 2. Preliminaries

Throughout this paper, let $[a,b] \subseteq \mathbb{R}$ be an interval and $0 < q < p \leq 1$ be a constant. The following definitions for the $(p,q)$-derivative and $(p,q)$-integral were given in [10,11].

**Definition 1.** *Let $f : [a,b] \to \mathbb{R}$ be a continuous function, and let $x \in [a,b]$. Then, the $(p,q)$-derivative of $f$ on $[a,b]$ at $x$ is defined as:*

$$_aD_{p,q}f(x) = \frac{f(px+(1-p)a) - f(qx+(1-q)a)}{(p-q)(x-a)}, \quad x \neq a \tag{3}$$

$$_aD_{p,q}f(a) = \lim_{x \to a} {_aD_{p,q}f(x)}.$$

Obviously, a function $f$ is $(p,q)$-differentiable on $[a,b]$ if $_aD_{p,q}f(x)$ exists for all $x \in [a,b]$. In Definition 1, if $a=0$, then $_0D_{p,q}f = D_{p,q}f$, where $D_{p,q}f$ is defined by:

$$D_{p,q}f(x) = \frac{f(px)-f(qx)}{(p-q)x}, \quad x \neq 0. \tag{4}$$

Furthermore, if $p=1$ in (4), then it reduces to $D_q f$, which is the $q$-derivative of the function $f$; see [5].

**Example 1.** *Define function $f : [a,b] \to \mathbb{R}$ by $f(x) = x^2 + 1$. Let $0 < q < p \leq 1$. Then, for $x \neq a$, we have:*

$$\begin{aligned}
_aD_{p,q}(x^2+1) &= \frac{\left[(px+(1-p)a)^2+1\right]-\left[(qx+(1-q)a)^2+1\right]}{(p-q)(x-a)} \\
&= \frac{(p+q)x^2+2ax[1-(p+q)]+a^2[(p+q)-2]}{(x-a)} \\
&= \frac{x(p+q)(x-a)-a(p+q)(x-a)+2a(x-a)}{(x-a)} \\
&= (p+q)(x-a)+2a.
\end{aligned} \tag{5}$$

**Definition 2.** *Let $f : [a,b] \to \mathbb{R}$ be a continuous function. Then, the $(p,q)$-integral on $[a,b]$ is defined by:*

$$\int_a^x f(t)\, _ad_{p,q}t = (p-q)(x-a)\sum_{n=0}^{\infty} \frac{q^n}{p^{n+1}} f\left(\frac{q^n}{p^{n+1}}x + \left(1 - \frac{q^n}{p^{n+1}}\right)a\right), \tag{6}$$

*for $x \in [a,b]$. If $a=0$ and $p=1$ in (6), then we have the classical $q$-integral [5].*

**Example 2.** *Define function $f : [a, b] \to \mathbb{R}$ by $f(x) = 2x$. Let $0 < q < p \le 1$. Then, we have:*

$$
\begin{aligned}
\int_a^b f(x) \, {}_a d_{p,q} x &= \int_a^b 2x \, {}_a d_{p,q} x \\
&= 2(p-q)(b-a) \sum_{n=0}^{\infty} \frac{q^n}{p^{n+1}} \left( \frac{q^n}{p^{n+1}} b + \left( 1 - \frac{q^n}{p^{n+1}} \right) a \right) \\
&= \frac{2(b-a)(b-a(1-p-q))}{p+q}.
\end{aligned}
\tag{7}
$$

**Theorem 1.** *Let $f : [a, b] \to \mathbb{R}$ be a continuous function. Then, we have the following:*

(i)　${}_a D_{p,q} \int_a^x f(t) \, {}_a d_{p,q} t = f(x)$;
(ii)　$\int_c^x {}_a D_{p,q} f(t) \, {}_a d_{p,q} t = f(x) - f(c)$ *for $c \in (a, x)$.*

**Theorem 2.** *Let $f, g : [a, b] \to \mathbb{R}$ be continuous functions and $\alpha \in \mathbb{R}$. Then, we have the following:*

(i)　$\int_a^x [f(t) + g(t)] \, {}_a d_{p,q} t = \int_a^x f(t) \, {}_a d_{p,q} t + \int_a^x g(t) \, {}_a d_{p,q} t$;
(ii)　$\int_a^x (\alpha f)(t) \, {}_a d_{p,q} t = \alpha \int_a^x f(t) \, {}_a d_{p,q} t$;
(iii)　$\int_c^x f(pt + (1-p)a) \, {}_a D_{p,q} g(t) \, {}_a d_q t = (fg)|_c^x - \int_c^x g(qt + (1-q)a) \, {}_a D_{p,q} f(t) \, {}_a d_{p,q} t$ *for $c \in (a, x)$.*

　　For the proof properties of Theorems 1 and 2, we refer to [10,11].
　　The proofs of the following theorems were given in [22].

**Theorem 3.** *Let $f : [a, b] \to \mathbb{R}$ be a convex differentiable function on $(a, b)$ and $0 < q < p \le 1$. Then, we have:*

$$
f \left( \frac{qa + pb}{p+q} \right) \le \frac{1}{p(b-a)} \int_a^{pb+(1-p)a} f(x) \, {}_a d_{p,q} x \le \frac{qf(a) + pf(b)}{p+q}.
\tag{8}
$$

**Theorem 4.** *Let $f : [a, b] \to \mathbb{R}$ be a convex differentiable function on $(a, b)$ and $0 < q < p \le 1$. Then, we have:*

$$
\begin{aligned}
f \left( \frac{pa+qb}{p+q} \right) + \frac{(p-q)(b-a)}{p+q} f' \left( \frac{pa+qb}{p+q} \right) &\le \frac{1}{p(b-a)} \int_a^{pb+(1-p)a} f(x) \, {}_a d_{p,q} x \\
&\le \frac{qf(a) + pf(b)}{p+q}.
\end{aligned}
\tag{9}
$$

**Theorem 5.** *Let $f : [a, b] \to \mathbb{R}$ be a convex differentiable function on $(a, b)$ and $0 < q < p \le 1$. Then, we have:*

$$
\begin{aligned}
f \left( \frac{a+b}{2} \right) + \frac{(p-q)(b-a)}{2(p+q)} f' \left( \frac{a+b}{2} \right) &\le \frac{1}{p(b-a)} \int_a^{pb+(1-p)a} f(x) \, {}_a d_{p,q} x \\
&\le \frac{qf(a) + pf(b)}{p+q}.
\end{aligned}
\tag{10}
$$

**Lemma 1.** *Let $f : [a, b] \to \mathbb{R}$ be a convex continuous function on $[a, b]$ and $0 < q < p \le 1$. Then, we have:*

$$
\begin{aligned}
&f \left( \frac{1}{(pb-pa)^2} \int_a^{pb+(1-p)a} \int_a^{pb+(1-p)a} (tx + (1-t)y) \, {}_a d_{p,q} x \, {}_a d_{p,q} y \right) \\
&\le \frac{1}{(pb-pa)^2} \int_a^{pb+(1-p)a} \int_a^{pb+(1-p)a} f(tx + (1-t)y) \, {}_a d_{p,q} x \, {}_a d_{p,q} y.
\end{aligned}
\tag{11}
$$

**Proof.** The proof of this lemma can be obtained by Definition 2 and Jensen's inequality.　□

## 3. Main Results

　　In this section, we present the $(p, q)$-Hermite–Hadamard inequality for double integrals and the refinement of Hermite–Hadamard inequalities on the interval $[a, b]$.

**Theorem 6.** *Let $f : [a, b] \to \mathbb{R}$ be a convex continuous function on $[a, b]$ and $0 < q < p \leq 1$. Then, we have:*

$$f\left(\frac{qa+pb}{p+q}\right) \leq \frac{1}{(pb-pa)^2} \int_a^{pb+(1-p)a} \int_a^{pb+(1-p)a} f(tx+(1-t)y) \; {}_a d_{p,q} x \, {}_a d_{p,q} y$$

$$\leq \frac{1}{p(b-a)} \int_a^{pb+(1-p)a} f(x) \; {}_a d_{p,q} x \, {}_a d_{p,q} y \tag{12}$$

$$\leq \frac{qf(a)+pf(b)}{p+q}.$$

**Proof.** Since $f$ is convex on $[a, b]$, it follows that:

$$f(tx+(1-t)y) \leq tf(x) + (1-t)f(y) \tag{13}$$

for all $x, y \in [a, b]$ and $t \in [0, 1]$. Taking the double $(p, q)$-integration on both sides for (13) on $[a, pb+(1-p)a] \times [a, pb+(1-p)a]$, we obtain:

$$\int_a^{pb+(1-p)a} \int_a^{pb+(1-p)a} f(tx+(1-t)y) \; {}_a d_{p,q} x \, {}_a d_{p,q} y$$

$$\leq \int_a^{pb+(1-p)a} \int_a^{pb+(1-p)a} [tf(x) + (1-t)f(y)] \; {}_a d_{p,q} x \, {}_a d_{p,q} y \tag{14}$$

$$= (pb-pa) \int_a^{pb+(1-p)a} f(x) \, {}_a d_{p,q} x,$$

which show the second part of (12) by using the right-hand side of the $(p, q)$-Hermite–Hadamard's inequality.

On the other hand, by Lemma 1, we have:

$$f\left(\frac{1}{(pb-pa)^2} \int_a^{pb+(1-p)a} \int_a^{pb+(1-p)a} (tx+(1-t)y) \; {}_a d_{p,q} x \, {}_a d_{p,q} y\right)$$

$$\leq \frac{1}{(pb-pa)^2} \int_a^{pb+(1-p)a} \int_a^{pb+(1-p)a} f(tx+(1-t)y) \; {}_a d_{p,q} x \, {}_a d_{p,q} y,$$

and since:

$$\frac{1}{(pb-pa)^2} \int_a^{pb+(1-p)a} \int_a^{pb+(1-p)a} (tx+(1-t)y) \; {}_a d_{p,q} x \, {}_a d_{p,q} y = \frac{qa+pb}{p+q}.$$

This completes the proof. □

**Remark 1.** *If $p = 1$ and $q \to 1$, then (12) reduces to (2), that is,*

$$f\left(\frac{a+b}{2}\right) \leq \frac{1}{(b-a)^2} \int_a^b \int_a^b f(tx+(1-t)y)dxdy \leq \frac{f(a)+f(b)}{2}.$$

**Corollary 1.** *Let $f : [a, b] \to \mathbb{R}$ be a convex continuous function on $[a, b]$ and $0 < q < p \leq 1$. Then, we have:*

$$f\left(\frac{qa+pb}{p+q}\right) \leq \frac{1}{(pb-pa)^2} \int_a^{pb+(1-p)a} \int_a^{pb+(1-p)a} f\left(\frac{x+y}{2}\right) \; {}_a d_{p,q} x \, {}_a d_{p,q} y$$

$$\leq \frac{1}{p(b-a)} \int_a^{pb+(1-p)a} f(x) \; {}_a d_{p,q} x$$

$$\leq \frac{qf(a)+pf(b)}{p+q}. \tag{15}$$

**Remark 2.** *If $p = 1$ and $q \to 1$, then (15) reduces to:*

$$f\left(\frac{a+b}{2}\right) \le \frac{1}{(b-a)^2} \int_a^b \int_a^b f\left(\frac{x+y}{2}\right) dxdy \le \frac{1}{b-a}\int_a^b f(x)dx \le \frac{f(a)+f(b)}{2},$$

*which readily appeared in [25].*

**Theorem 7.** *Let $f : [a,b] \to \mathbb{R}$ be a convex continuous function on $[a,b]$ and $0 < q < p \le 1$. Then, we have:*

$$\frac{p}{(pb-pa)^2} \int_a^{pb+(1-p)a} \int_a^{pb+(1-p)a} f\left(\frac{px+qy}{p+q}\right) {}_a d_{p,q}x {}_a d_{p,q}y$$

$$\le \frac{1}{(pb-pa)^2}\int_0^p \int_a^{pb+(1-p)a}\int_a^{pb+(1-p)a} f(tx+(1-t)y) {}_a d_{p,q}x {}_a d_{p,q}y {}_a d_{p,q}t \qquad (16)$$

$$\le \frac{1}{pb-pa}\int_a^{pb+(1-p)a} f(x) {}_a d_{p,q}x.$$

**Proof.** Let $g : [a,b] \to \mathbb{R}$ be given by:

$$g(t) = \frac{1}{(pb-pa)^2}\int_a^{pb+(1-p)a}\int_a^{pb+(1-p)a} f(tx+(1-t)y) {}_a d_{p,q}x {}_a d_{p,q}y.$$

For all $t_1, t_2 \in [0,1]$ and $\alpha, \beta \ge 0$ with $\alpha + \beta = 1$, we consider:

$$g(\alpha t_1 + \beta t_2) = \frac{1}{(pb-pa)^2}\int_a^{pb+(1-p)a}\int_a^{pb+(1-p)a} f((\alpha t_1+\beta t_2)x + (1-(\alpha t_1 + \beta t_2))y) {}_a d_{p,q}x {}_a d_{p,q}y$$

$$\le \frac{\alpha}{(pb-pa)^2}\int_a^{pb+(1-p)a}\int_a^{pb+(1-p)a} f(t_1 x + (1-t_1)y) {}_a d_{p,q}x {}_a d_{p,q}y$$

$$+ \frac{\beta}{(pb-pa)^2}\int_a^{pb+(1-p)a}\int_a^{pb+(1-p)a} f(t_2 x + (1-t_2)y) {}_a d_{p,q}x {}_a d_{p,q}y$$

$$= \alpha g(t_1) + \beta g(t_2),$$

which show that $g$ is convex on $[0,1]$. Using Theorem 3 for the convex function $g$, we have:

$$\frac{1}{(pb-pa)^2}\int_a^{pb+(1-p)a}\int_a^{pb+(1-p)a} f\left(\frac{px+qy}{p+q}\right)_a d_q x_a d_q y$$

$$= g\left(\frac{p}{p+q}\right) \le \frac{1}{p}\int_0^p g(t) {}_a d_{p,q}t$$

$$= \frac{1}{p(pb-pa)^2}\int_0^p \int_a^{pb+(1-p)a}\int_a^{pb+(1-p)a} f(tx+(1-t)y) {}_a d_{p,q}x {}_a d_{p,q}y {}_a d_{p,q}t$$

$$\le \frac{qg(0)+pg(1)}{p(p+q)} = \frac{1}{p(pb-pa)}\int_a^{pb+(1-p)a} f(x) {}_a d_{p,q}x.$$

This completes the proof. □

**Remark 3.** *If $p = 1$ and $q \to 1$, then (16) reduces to:*

$$\frac{1}{(b-a)^2}\int_a^b\int_a^b f\left(\frac{x+y}{2}\right)dxdy \le \frac{1}{(b-a)^2}\int_0^1\int_a^b\int_a^b f(tx+(1-t)y)dxdydt$$

$$\le \frac{1}{b-a}\int_a^b f(x)dx,$$

*which readily appeared in [25].*

**Theorem 8.** *Let $f : [a, b] \to \mathbb{R}$ be a $(p, q)$-differentiable convex continuous function and $0 < q < p \leq 1$, then the following inequalities:*

$$
\begin{aligned}
0 \quad &\leq \frac{p}{b-a} \int_a^{pb+(1-p)a} f(x) \, _a d_{p,q} x \\
&- \frac{1}{(b-a)^2} \int_a^{pb+(1-p)a} \int_a^{pb+(1-p)a} f(tx + (1-t)y) \, _a d_{p,q} x _a d_{p,q} y \\
&\leq t \left[ \frac{p^2 f(a) + pqf(pb+(1-p)a)}{p+q} - \frac{p}{b-a} \int_a^{pb+(1-p)a} f(qx + (1-q)a) \, _a d_{p,q} x \right],
\end{aligned}
\tag{17}
$$

*are valid for all $t \in [0, 1]$.*

**Proof.** Since $f$ is convex on $J$, it follows that:

$$
f(tx + (1-t)y) \leq tf(x) + (1-t)f(y)
$$

for all $x, y \in [a, b]$ and $t \in [0, 1]$. Taking double $(p, q)$-integration on both sides of the above inequality on $[a, pb + (1-p)a] \times [a, pb + (1-p)a]$, we obtain:

$$
\begin{aligned}
\int_a^{pb+(1-p)a} &\int_a^{pb+(1-p)a} f(tx + (1-t)y) \, _a d_{p,q} x _a d_{p,q} y \\
&\leq \int_a^{pb+(1-p)a} \int_a^{pb+(1-p)a} [tf(x) + (1-t)f(y)] \, _a d_{p,q} x _a d_{p,q} y \\
&= p(b-a) \int_a^{pb+(1-p)a} f(x) _a d_{p,q} x.
\end{aligned}
$$

On the other hand, since $f$ is $(p, q)$-differentiable convex on $[a, b]$ and $f' \geq \, _a D_{p,q} f$, we have:

$$
f(tx + (1-t)y) - f(y) \geq t(x - y) \, _a D_{p,q} f(y)
$$

for all $x, y \in [a, b]$ and $t \in [0, 1]$. Taking the double $(p, q)$-integration on both sides of the above inequality on $[a, pb + (1-p)a] \times [a, pb + (1-p)a]$, we obtain:

$$
\begin{aligned}
\int_a^{pb+(1-p)a} &\int_a^{pb+(1-p)a} f(tx + (1-t)y) \, _a d_{p,q} x _a d_{p,q} y - (pb - pa) \int_a^{pb+(1-p)a} f(x) \, _a d_{p,q} x \\
&\geq t \int_a^{pb+(1-p)a} \int_a^{pb+(1-p)a} (x - y) _a D_{p,q} f(y) \, _a d_{p,q} x _a d_{p,q} y.
\end{aligned}
\tag{18}
$$

Since,

$$
\begin{aligned}
\int_a^{pb+(1-p)a} &\int_a^{pb+(1-p)a} (x - y) _a D_{p,q} f(y) \, _a d_{p,q} x _a d_{p,q} y \\
&= (pb - pa) \int_a^{pb+(1-p)a} f(qx + (1-q)a) \, _a d_{p,q} x - (b - a)^2 \frac{[p^2 f(a) + pqf(pb + (1 - p)a)]}{p + q}.
\end{aligned}
$$

Substituting the above inequality in (18), we have:

$$
\begin{aligned}
(pb - pa) \int_a^{pb+(1-p)a} &f(x) \, _a d_{p,q} x - \int_a^{pb+(1-p)a} \int_a^{pb+(1-p)a} f(tx + (1-t)y) \, _a d_{p,q} x _a d_{p,q} y \\
&\leq t \left[ (b - a)^2 \frac{[p^2 f(a) + pqf(pb + (1 - p)a)]}{p + q} - (pb - pa) \int_a^{pb+(1-p)a} f(qx + (1-q)a) \, _a d_{p,q} x \right]
\end{aligned}
$$

for all $t \in [0, 1]$, which completes the proof. $\square$

**Remark 4.** *If $p = 1$ and $q \to 1$, then* (17) *reduces to:*

$$0 \le \frac{1}{b-a} \int_a^b f(x)dx - \frac{1}{(b-a)^2} \int_a^b \int_a^b f(tx + (1-t)y)dxdy$$

$$\le t \left[ \frac{f(a) + f(b)}{2} - \frac{1}{b-a} \int_a^b f(x)dx \right],$$

*which readily appeared in* [25,26].

**Corollary 2.** *Let $f : [a, b] \to \mathbb{R}$ be a $(p, q)$-differentiable convex continuous function and $0 < q < p \le 1$. Then, we have:*

$$0 \le \frac{p}{b-a} \int_a^{pb+(1-p)a} f(x) \,_a d_{p,q}x - \frac{1}{(b-a)^2} \int_a^{pb+(1-p)a} \int_a^{pb+(1-p)a} f\left(\frac{x+y}{2}\right) \,_a d_{p,q}x \,_a d_{p,q}y$$

$$\le \frac{1}{2} \left[ \frac{p^2 f(a) + pqf(pb+(1-p)a)}{p+q} - \frac{p}{b-a} \int_a^{pb+(1-p)a} f(qx + (1-q)a) \,_a d_{p,q}x \right]. \tag{19}$$

**Remark 5.** *If $p = 1$ and $q \to 1$, then* (19) *reduces to:*

$$0 \le \frac{1}{b-a} \int_a^b f(x)dx - \frac{1}{(b-a)^2} \int_a^b \int_a^b f\left(\frac{x+y}{2}\right)dxdy$$

$$\le \frac{1}{2} \left[ \frac{f(a) + f(b)}{2} - \frac{1}{b-a} \int_a^b f(x)dx \right],$$

*which readily appeared in* [25].

**Theorem 9.** *Let $f : [a, b] \to \mathbb{R}$ be a $(p, q)$-differentiable convex continuous function, which is defined at the point $\frac{qa+pb}{p+q} \in (a, b)$ and $0 < q < p \le 1$. Then, the following inequalities:*

$$0 \le \frac{1}{b-a} \int_a^{pb+(1-p)a} f(x) \,_a d_{p,q}x - \frac{1}{b-a} \int_a^{pb+(1-p)a} f\left(tx + (1-t)\frac{qa+pb}{p+q}\right) \,_a d_{p,q}x$$

$$\le (1-t) \left[ \frac{pf(a) + qf(pb+(1-p)a)}{p+q} - \frac{1}{b-a} \int_a^{pb+(1-p)a} f(qx + (1-q)a) \,_a d_{p,q}x \right] \tag{20}$$

*are valid for all $t \in [0, 1]$.*

**Proof.** Since $f$ is convex on $[a, b]$ and using Theorem 3, we have:

$$\frac{1}{p(b-a)} \int_a^{pb+(1-p)a} f\left(tx + (1-t)\frac{qa+pb}{p+q}\right) \,_a d_{p,q}x$$

$$\le \frac{t}{p(b-a)} \int_a^{pb+(1-p)a} f(x) \,_a d_{p,q}x + (1-t)f\left(\frac{qa+pb}{p+q}\right)$$

$$\le \frac{t}{p(b-a)} \int_a^{pb+(1-p)a} f(x) \,_a d_{p,q}x + \frac{1-t}{p(b-a)} \int_a^{pb+(1-p)a} f(x) \,_a d_{p,q}x$$

$$= \frac{1}{p(b-a)} \int_a^{pb+(1-p)a} f(x) \,_a d_{p,q}x$$

for all $t \in [0, 1]$.

On the other hand, since $f$ is the $(p, q)$-differentiable convex on $[a, b]$, we have:

$$f\left(tx + (1-t)\frac{qa+pb}{p+q}\right) - f(x) \ge (1-t)\left(\frac{qa+pb}{p+q} - x\right) \,_a D_{p,q}(x).$$

Taking the double $(p,q)$-integration on both sides of the above inequality on $[a,b]$, we obtain:

$$\frac{1}{p(b-a)} \int_a^{pb+(1-p)a} f\left(tx + (1-t)\frac{qa+pb}{p+q}\right) \, _a d_{p,q}x - \frac{1}{p(b-a)} \int_a^{pb+(1-p)a} f(x) \, _a d_{p,q}x$$
$$\geq \frac{(1-t)}{p(b-a)} \int_a^{pb+(1-p)a} \left(\frac{qa+pb}{p+q} - x\right) \, _a D_{p,q} f(x)_a d_{p,q}x. \tag{21}$$

Since,

$$\int_a^{pb+(1-p)a} \left(\frac{qa+pb}{p+q} - x\right) \, _a D_{p,q} f(x)_a d_{p,q}x$$
$$= \int_a^{pb+(1-p)a} f(qx + (1-q)a)_a d_{p,q}x - (b-a)\frac{pf(a)+qf(pb+(1-p)a)}{p+q}. \tag{22}$$

This completes the proof.  □

**Corollary 3.** *Let $f : [a,b] \to \mathbb{R}$ be a $(p,q)$-differentiable convex continuous function and $0 < q < p \leq 1$. Then, we have:*

$$0 \leq \frac{1}{b-a} \int_a^{pb+(1-p)a} f(x) \, _a d_{p,q}x - \frac{2}{b-a} \int_{\frac{a(p+2q)+pb}{2(1+q)}}^{\frac{(p^2+pq)(b-a)+(p+2q)a+pb}{2(p+q)}} f(x) \, _a d_{p,q}x$$
$$\leq \frac{1}{2}\left[\frac{pf(a)+qf(pb+(1-p)a)}{p+q} - \frac{1}{b-a} \int_a^{pb+(1-p)a} f(qx + (1-q)a) \, _a d_{p,q}x\right]. \tag{23}$$

**Theorem 10.** *Let $f : [a,b] \to \mathbb{R}$ be a $(p,q)$-differentiable convex continuous function, which is defined at the point $\frac{pa+qb}{p+q} \in (a,b)$ and $0 < q < p \leq 1$. Then, the following inequalities:*

$$(1-t)\frac{p(p-q)(b-a)}{p+q} f'\left(\frac{pa+qb}{p+q}\right)$$
$$\leq \frac{1}{b-a} \int_a^{pb+(1-p)a} f(x) \, _a d_{p,q}x - \frac{1}{b-a} \int_a^{pb+(1-p)a} f\left(tx + (1-t)\frac{pa+qb}{p+q}\right) \, _a d_{p,q}x \tag{24}$$
$$\leq (1-t)\left[\frac{qf(a)+pf(pb+(1-p)a)}{p+q} - \frac{1}{b-a} \int_a^{pb+(1-p)a} f(qx + (1-q)a) \, _a d_{p,q}x\right]$$

*are valid for all $t \in [0,1]$.*

**Proof.** The proof of this theorem follows a similar procedure as Theorem 9 by using Theorem 4.  □

**Corollary 4.** *Let $f : [a,b] \to \mathbb{R}$ be a $(p,q)$-differentiable convex continuous function and $0 < q < p \leq 1$. Then, we have:*

$$\frac{p(p-q)(b-a)}{2(p+q)} f'\left(\frac{pa+qb}{p+q}\right)$$
$$\leq \frac{1}{b-a} \int_a^{pb+(1-p)a} f(x) \, _a d_{p,q}x - \frac{2}{b-a} \int_{\frac{2pa+q(a+b)}{2(p+q)}}^{\frac{(p^2+pq)(b-a)+(2p+q)a+qb}{2(1+q)}} f(x) \, _a d_{p,q}x \tag{25}$$
$$\leq \frac{1}{2}\left[\frac{qf(a)+pf(pb+(1-p)a)}{p+q} - \frac{1}{b-a} \int_a^{pb+(1-p)a} f(qx + (1-q)a) \, _a d_{p,q}x\right].$$

**Theorem 11.** *Let $f : [a,b] \to \mathbb{R}$ be a $(p,q)$-differentiable convex continuous function, which is defined at the point $\frac{a+b}{2} \in (a,b)$ and $0 < q < p \leq 1$. Then, the following inequalities:*

$$(1-t)\frac{p(p-q)(b-a)}{2(p+q)} f'\left(\frac{a+b}{2}\right)$$
$$\leq \frac{1}{b-a} \int_a^{pb+(1-p)a} f(x) \, _a d_{p,q}x - \frac{1}{b-a} \int_a^{pb+(1-p)a} f\left(tx + (1-t)\frac{a+b}{2}\right) \, _a d_{p,q}x \tag{26}$$
$$\leq (1-t)\left[\frac{f(a)+f(pb+(1-q)a)}{2} - \frac{1}{b-a} \int_a^{pb+(1-p)a} f(qx + (1-q)a) \, _a d_{p,q}x\right]$$

*are valid for all $t \in [0,1]$.*

**Proof.** The proof of this theorem follows a similar procedure as Theorem 9 by using Theorem 5. □

**Corollary 5.** *Let $f : [a,b] \to \mathbb{R}$ be a $(p,q)$-differentiable convex continuous function and $0 < q < p \leq 1$. Then, we have:*

$$
\frac{p(p-q)(b-a)}{4(p+q)} f'\left(\frac{a+b}{2}\right)
$$

$$
\leq \frac{1}{b-a} \int_a^{pb+(1-p)a} f(x)\,_a d_{p,q}x - \frac{2}{b-a} \int_{\frac{3a+b}{4}}^{\frac{2p(b-a)+3a+b}{4}} f(x)\,_a d_{p,q}x \tag{27}
$$

$$
\leq \frac{1}{2}\left[\frac{qf(a)+pf(pb+(1-p)a)}{p+q} - \frac{1}{b-a}\int_a^{pb+(1-p)a} f(qx+(1-q)a)\,_a d_{p,q}x\right].
$$

**Remark 6.** *If $p=1$ and $q \to 1$, then (20), (24), and (26) reduce to:*

$$
0 \leq \frac{1}{b-a}\int_a^b f(x)dx - \frac{1}{b-a}\int_a^b f\left(tx+(1-t)\frac{a+b}{2}\right)dx
$$

$$
\leq (1-t)\left[\frac{f(a)+f(b)}{2} - \frac{1}{b-a}\int_a^b f(x)dx\right],
$$

*which readily appeared in [25].*

**Remark 7.** *If $p=1$ and $q \to 1$, then (23), (25), and (27) reduce to:*

$$
0 \leq \frac{1}{b-a}\int_a^b f(x)dx - \frac{2}{b-a}\int_{\frac{3a+b}{4}}^{\frac{a+3b}{4}} f(x)dx \leq \frac{1}{2}\left[\frac{f(a)+f(b)}{2} - \frac{1}{b-a}\int_a^b f(x)dx\right],
$$

*which readily appeared in [25].*

## 4. Conclusions

In this paper, we have obtained some new results for the $(p,q)$-calculus of Hermite–Hadamard inequalities for the double integral and refinements of the Hermite–Hadamard inequality. Our work has improved the results of [23] and can be reduced to the classical inequality formulas in special cases when $p=1$ and $q \to 1$. It is expected that this paper may stimulate further research in this field.

**Author Contributions:** The order of the author list reflects the contributions to the paper.

**Funding:** This research received no external funding.

**Conflicts of Interest:** The authors declare no conflict of interest.

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
