# Peer review of "(p, q)-Hermite–Hadamard Inequalities for Double Integral and (p, q)-Differentiable Convex Functions"

_axioms, doi:10.3390/axioms8020068_

Round 1

Reviewer 1 Report

The aim of the paper is to establish some well-known inequalities for the (p,q) calculus. The paper is therefore rather technical, but with the clear structure and easy to go through. I would just add a short conclusion paragraph about perspectives and possible applications.

Author Response

Dear the Reviewer,   We thank you for your careful reading of the manuscript and helpful comments  and suggestions. English writing has already been rechecked again in the new  revised manuscript by the authors.We would be grateful if the manuscript could be considered for  publication in Axioms.
Best regards,Kamsing  Nonlaopon

Reviewer 2 Report

Dear Authors,

the paper has sufficient scientific merit and the contribution to the existing knowledge. However please present any valuable conslusions based on the obtained solutions.

Author Response

Dear the reviewer,      We thank you for your careful reading of the manuscript and helpful  comments and suggestions. English writing has already been rechecked again  in the new revised manuscript by the authors.We would be grateful if the manuscript could be considered for  publication in Axioms.
Best regards,Kamsing  Nonlaopon

Reviewer 3 Report

You need to detail the proof of Lemma 1.

Justify the inequality from Theorem 8: the derivative of f is greater than (p,q)-derivative of f.

Author Response

Dear reviewer,

We thank you for your careful reading of the manuscript and helpful  comments and suggestions. We have made revisions according to your comments  and suggestions in attached file. We would be grateful if the manuscript could be considered for  publication in Axioms. Best regards, Kamsing  Nonlaopon

Reviewer 4 Report

The authors prove some inequalities.

Initially  they give a definition of (p,q) derivative and integral.  Since these definitions are not authors’ ones, they have to cite.

About definition 2- Is there a restriction about p and q in Defintion 2. Is it assumed the inequality written on line 38  page 2 is satisfied? If not, then it has to be written. If yes, then the proof of Theorem 6  is not correct. Also, for what p and q Theorem 1 is satisfied?

How could Corollary 1 be obtained from Theorem 6? Please, see the restrictions about p and q.

The first 4 lines in the proof of Theorem 7 have  to be corrected.

Finally, to be given at least one example illustrating some of the obtained inequalities.

Author Response

(The authors gave the same response as above.)

Round 2

Reviewer 4 Report

I'm not satisfied by the authors changes and answers. 

The main complain is still actual:

The authors added line 38(Incorrect English) . Now, look at the conditions of Theorem 6. They contradict the added lines. Also, they don't allow Corollary 1 to be obtained from Theorem 6.  Also, the given by authors example does not satisfy the conditions of Theorem 6.

I would like to see at least one function (different than the tricial f(x)=x which is satisfying the conditions.

Author Response

Dear the reviewer,We thank you for your careful reading of the manuscript and helpful comments  and suggestions. We have made revisions according to your comments and  suggestions, as described in attached file.    Best regards, Dr.Kamsing Nonlaopon

Round 3

Reviewer 4 Report

OK, now the paper is fine.